# A neurovascular high-frequency optical coherence tomography system enables in situ cerebrovascular volumetric microscopy

Giovanni J. Ughi [1,2,9], Miklos G. Marosfoi [1,8,9], Robert M. King [1,3], Jildaz Caroff [1,4], Lindsy M. Peterson[2], Benjamin H. Duncan [2], Erin T. Langan[1], Amanda Collins [5], Anita Leporati[1], Serge Rousselle[6], Demetrius K. Lopes [7], Matthew J. Gounis [1✉] & Ajit S. Puri[1]

Intravascular imaging has emerged as a valuable tool for the treatment of coronary and peripheral artery disease; however, no solution is available for safe and reliable use in the tortuous vascular anatomy of the brain. Endovascular treatment of stroke is delivered under image guidance with insufficient resolution to adequately assess underlying arterial pathology and therapeutic devices. High-resolution imaging, enabling surgeons to visualize cerebral arteries' microstructure and micron-level features of neurovascular devices, would have a profound impact in the research, diagnosis, and treatment of cerebrovascular diseases. Here, we present a neurovascular high-frequency optical coherence tomography (HF-OCT) system, including an imaging console and an endoscopic probe designed to rapidly acquire volumetric microscopy data at a resolution approaching 10 microns in tortuous cerebrovascular anatomies. Using a combination of in vitro, ex vivo, and in vivo models, the feasibility of HF-OCT for cerebrovascular imaging was demonstrated.

[1] New England Center for Stroke Research, Department of Radiology, University of Massachusetts Medical School, Worcester, MA, USA. [2] Gentuity LLC, Sudbury, MA, USA. [3] Department of Biomedical Engineering, Worcester Polytechnic Institute, Worcester, MA, USA. [4] Department of Interventional Neuroradiology, NEURI Center, Bicêtre Hospital, Le Kremlin-Bicêtre, France. [5] Division of Translational Anatomy, Department of Radiology, University of Massachusetts Medical School, Worcester, MA, USA. [6] Alizée Pathology, Inc, Thurmont, MD, USA. [7] Department of Neurosurgery, Advocate Health, Chicago, IL, USA. [8] Present address: Department of Neurointerventional Radiology, Beth Israel Lahey Clinic, Burlington, MA, USA. [9] These authors contributed equally: Giovanni J. Ughi, Miklos G. Marosfoi. ✉email: Matthew.Gounis@umassmed.edu

Despite tremendous advances in minimally invasive therapies, there remain limitations to endovascular treatments of brain arteries and aneurysms, largely related to inadequate visualization techniques. Non-invasive imaging technologies are unable to provide sufficient resolution to adequately assess the underlying artery pathology, small perforating arteries, the device–vessel relationship, and device-related effects such as platelet aggregation. Intravascular (IV)-imaging solutions are available for the imaging of coronary and peripheral arteries; however, they are unsuitable for routine cerebrovascular use.

In the last decade, newer generations of endovascular devices designed for the cerebrovasculature, such as self-expanding microstents and flow diverters with struts as small as 25 μm, have enabled the treatment of wide-neck, complex brain aneurysms[1]. Given the limited x-ray attenuation of these devices, radiopaque markers are distributed along the devices to provide basic fluoroscopy location guidance; however, it is currently not possible to image the device in its entirety. This is critically important since an accurate device placement with respect to the pathology and apposition are necessary for effective treatment and prevention of disabling complications[2,3]. Neurovascular devices require particularly precise placement of the device to achieve the ultimate goal that is complete exclusion of the aneurysm from the circulation, and even when angiographic satisfactory results are achieved acutely, long-term complication rates remain high[4].

Similarly, endovascular thrombectomy has become the standard of care for eligible patients suffering from ischemic stroke due to a large vessel occlusion[5]. Non-invasive imaging techniques such as magnetic resonance vessel wall imaging have been proposed to assess vascular damage following thrombectomy[6] or to detect underlying pathology such as intracranial atherosclerosis[7]; however, they have an insufficient resolution to directly visualize the vascular pathology (e.g., intracranial atherosclerotic disease or dissection), endothelial injury, or thrombosis of perforating arteries.

Volumetric microscopy with the ability to visualize neurovascular devices in their entirety and the vessel wall microstructure in vivo may have a profound impact in endovascular neurosurgery[8–10]. Endoscopic optical-imaging techniques, with a resolution approaching the micron scale and feasible incorporation in small fiber-optics probes, are a promising candidate for a successful translation to the clinic[11,12]. In the last decade, IV optical coherence tomography (OCT) or optical frequency domain imaging (OFDI) has become an increasingly popular modality, as it enables accurate measurements of the coronary lumen morphology, and disease severity[13–16]. Furthermore, OCT/OFDI has been extensively used to investigate newer generations of intracoronary stent devices[17–20]. Beyond coronary applications, IV imaging has the potential to revolutionize the diagnosis and management of cerebrovascular pathologies[21,22]; however, until now, commercial imaging catheters lack the flexibility required to be advanced in tortuous cerebrovascular anatomy, are not designed to perform imaging in elevated tortuosity, and are incompatible with the neurovascular clinical workflow[21]. As such, the use of IV imaging in the clinical setting has been restricted to the posterior circulation of a few selected patients with limited tortuosity[23] and to proximal, non-tortuous segments of internal carotid arteries[24,25]. In addition, the field of view (i.e., acquired image diameter) of the existing devices is insufficient to characterize large and complex carotid arteries with a diameter of 5 mm or more, and intracranial aneurysms. In this study, we introduce a high-frequency OCT (HF-OCT) imaging system, including an imaging console and a fiber-optic endoscopic probe designed for cerebrovascular use and to satisfy these requirements, thereby enabling cerebrovascular volumetric in vivo microscopy.

## Results

### A neurovascular high-resolution endoscopic imaging system.
We used prototypes of a HF-OCT imaging console and of an endoscopic probe designed for neurovascular applications to perform volumetric microscopy in a combination of in vitro, in vivo, and ex vivo models of cerebrovascular arteries and therapeutic devices. The term HF-OCT describes a long coherence length frequency domain OCT-imaging system (see "Methods" section) specifically designed for use in cerebrovascular anatomy. The probe introduced in this study is named Vis-M and is a flexible, 0.016-inch (~400 μm) OD wire-like catheter devised for navigation in tortuous intracranial anatomy. Contrary to existing IV imaging solutions, the Vis-M device does not require guidewire supported navigation, and can be delivered through standard, distal access neurovascular catheters. The Vis-M optics has the ability to focus and collect near-infrared light that is back-scattered from the vessel wall, neurovascular devices, and intraluminal objects. Through a rapid rotation of the imaging lens within the Vis-M protective sheath and by retracting the probe through the vessel, the HF-OCT system performs volumetric imaging of the surrounding artery and devices at an axial resolution approaching 10 μm (Fig. 1). The Vis-M is designed to have a comparable profile with state-of-the-art neurovascular guidewires and features an atraumatic, radiopaque tip. The catheter construct uses a distal rotational control solution and avoids the use of a torque cable, resulting in greatly improved device flexibility and decreased size (see "Methods" section). This solution enables a significantly reduced crossing profile, less than half of the diameter and less than a quarter of the cross-sectional area of commercial IV-imaging solutions[13]. These characteristics enable a smooth navigation and reliable image quality in elevated vascular tortuosity conditions, inaccessible with existing IVUS and IV OCT technologies[8]. The Vis-M inner optics rotates at a speed of 250 Hz, with each rotation corresponding to the acquisition of a cross-sectional frame, with a field of view capable of visualizing arteries with a diameter up to ~7 mm. This frame rate enables the collection of high-resolution, cross-sectional HF-OCT images from tortuous arterial segments with a longitudinal spacing of 80 μm, in the example of a 50 mm pullback acquired in 2 and a half seconds. Shorter or longer pullback lengths (up to 80 mm) and durations can be selected by the user to optimize the acquisition parameters for an arterial segment of a given length.

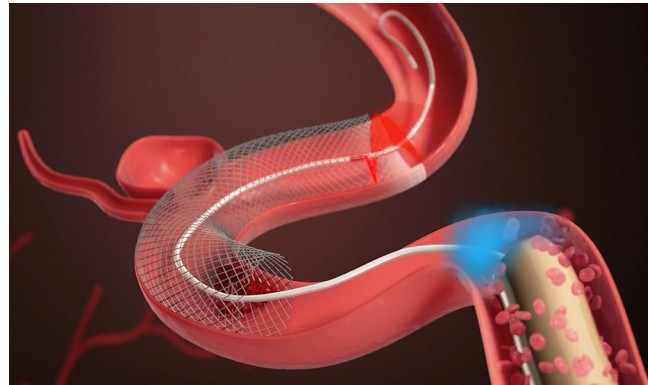

**Fig. 1 Principles of HF-OCT imaging in an intracranial vessel.** During a brief injection of contrast (blue color) through a 5 F distal access catheter, the Vis-M device is retracted while quickly rotating its internal optics resulting in a helical scanning pattern. By the means of a tightly spaced pattern (e.g., 80 μm) and an axial resolution approaching 10 μm, comprehensive volumetric microscopy of the arterial wall, neurovascular devices, and intraluminal objects is obtained.

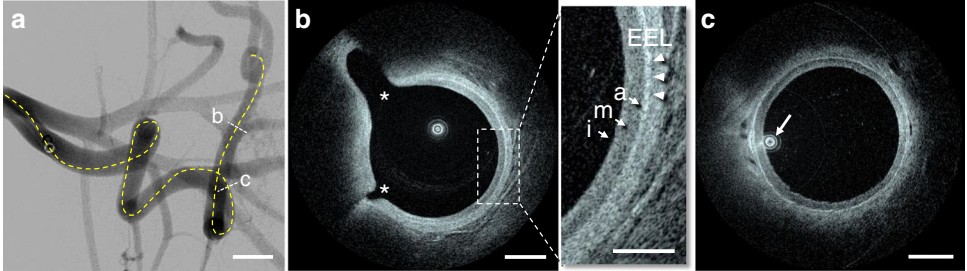

**Fig. 2 In vivo imaging in a flexed forelimb model of brachial porcine artery. a** The dotted line highlights the estimated path taken by the Vis-M device through the vessel tortuosity. **b** HF-OCT microscopy shows the external elastic lamina (arrowheads) and the individual layers of the vessel wall (arrows). A bright tunica intima is followed by a dark tunica media and a bright adventitia (inset). The asterisks denote the ostia of two side-branches, with diameters of ~0.2 and 0.7 mm, respectively. **c** The arrow indicates the eccentric position of the HF-OCT device in the arterial lumen. The image shows a uniform illumination and absence of NURD artifacts. Imaging in a flexed forelimb swine model was repeated in a total of $n = 16$ arteries from all animals included in this study ($n = 8$). The scale bar on DSA image is equal to 1 cm (**a**). HF-OCT images scale bars are equal to 1 mm (**b** and **c**), and 0.5 mm in the inset.

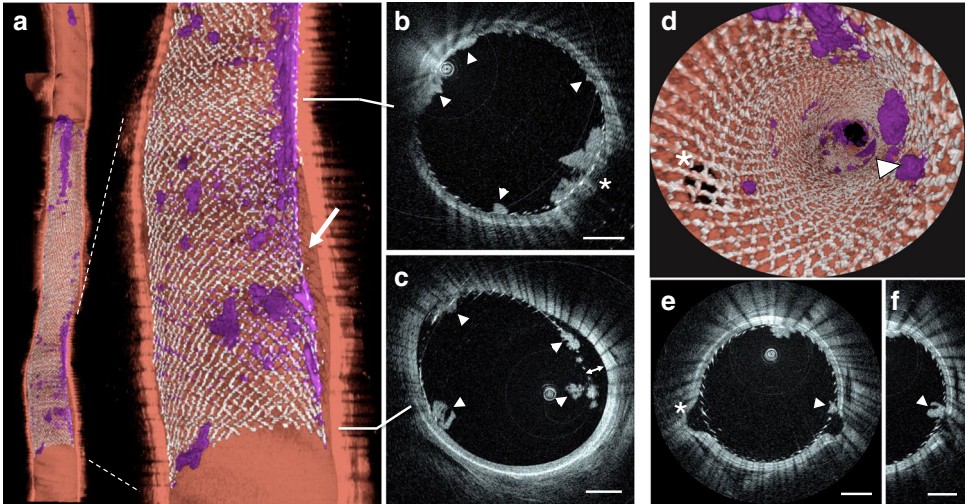

**Fig. 3 HF-OCT volumetric rendering of a stented swine internal maxillary artery. a** HF-OCT three-dimensional cutaway rendering (top = distal; bottom = proximal). Flow-diverter malapposition (arrow) and clots of different sizes (purple color) are visible over the flow-diverter surface. **b** Cross-sectional HF-OCT imaging show a jailed branch (asterisk) and several thrombus formations over the flow-diverter surface (arrowheads). **c** Flow-diverter proximal edge, with incomplete apposition (3 o'clock) and several clots over the device surface (arrowheads). **d** Endoscopic view of HF-OCT volumetric rendering. Small, perforator-like side-branches jailed by a flow-diverting stent are visible. **e** The side-branch located on the left side of the image (arrow) is free of clots and the flow-diverting stent is well-apposed to the parent artery. **f** A second branch located on the right (indicated by the asterisk) shows flow-diverter struts that are embedded by a clot. Stented arteries data collection was repeated in a total of $n = 16$ swine internal maxillary arteries, including $n = 16$ flow-diverting stents, and $n = 15$ self-expanding intracranial stents. Scale bars on all HF-OCT images are equal to 1 mm. Three-dimensional rendering color scheme: red, artery wall; purple, clot; silver, metallic struts.

**Vessel clearing in an in vitro model of the circle of Willis**. Red blood cells scatter light and degrade its coherence properties and, as such, the acquisition of HF-OCT data requires the displacement of arterial blood from the vessel lumen. To test the ability to infuse sufficient contrast agent for the creation of a suitable optical window, simulated use experiments were performed at the main anatomical locations in a patient-specific bench model of the complete circle of Willis (Supplementary Fig. 1). Omnipaque 350 was identified as the preferred media, having an optimal viscosity to efficiently displace the blood from the image field of view. Complete clearance of the internal carotid artery (ICA) was obtained by flushing at 5 ml/s and was observed after ~2–3 s of injection. Similarly, clearance of the middle cerebral artery (MCA) and the vertebral artery (VA) was obtained at a rate of 3 ml/s. The basilar artery (BA) required a 5 ml/s injection to overcome the inflow from both VAs with the 5 F catheter positioned at the level of the BA ostium. A rate of 4 ml/s may be able to provide sufficient clearance when injecting contrast within a more distal position of the BA[23]. These results indicate that HF-

OCT acquisitions with a duration of 2 s require injections for a minimum of 4 s to clear the artery, resulting in a 20 ml injection of contrast media for the ICA at 5 ml/s and a 12 ml injection for the MCA at 3 ml/s, similar to clinical protocols for three-dimensional angiography[26].

**Imaging in elevated vascular tortuosity in vivo**. A flexed forelimb swine model ($n = 8$) provided severe tortuosity conditions in brachial arteries, with a curvature similar to the human internal carotid artery siphon[27]. The tortuosity provided by this in vivo model cannot reliably be accessed and dependably imaged with existing IV imaging solutions[8]. The Vis-M device was tested bilaterally in brachial arteries ($n = 16$) acquiring HF-OCT data sets with an effective imaging length of $65 \pm 13$ mm. The performance of the Vis-M devices was evaluated by assessing image artifacts that typically affect state-of-the-art IV technologies in tortuous anatomy, such as non-uniform rotational distortion (NURD)[28]. This artifact occurs when increased friction is exerted on the rotating catheter components such as the torque cable;

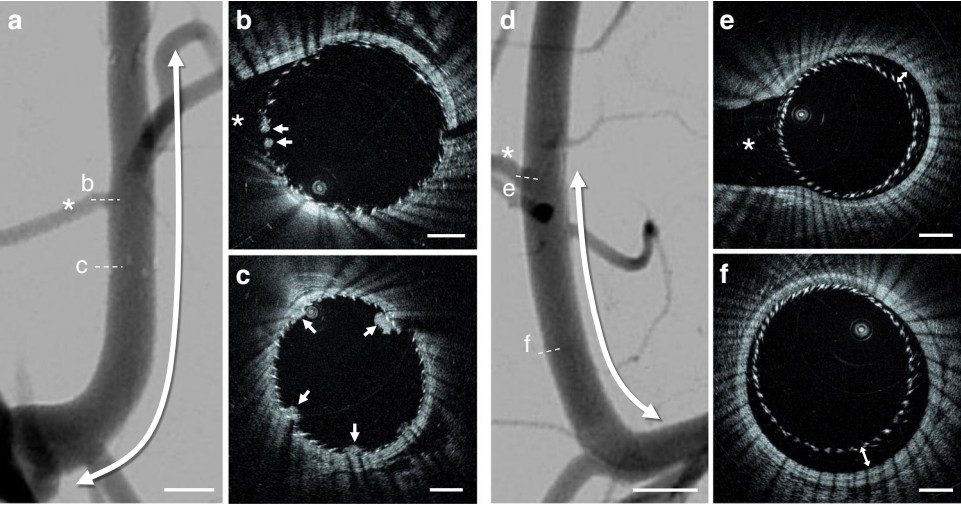

**Fig. 4 HF-OCT imaging comparison with DSA. a** Digital subtraction angiography (DSA) of an internal maxillary artery stented using flow-diverting and neurovascular stents. The stented segment is indicated by the arrow, with the devices partially overlapping. Corresponding HF-OCT images show thrombus formations (arrows) not observed on fluoroscopy located over the struts of the flow-diverter at the level of a side branch **b**, and over the struts of the flow-diverter and intra-cranial stent in the overlapping segment **c**. **d** DSA of a second IMAX artery treated with a flow-diverting stent. The corresponding HF-OCT cross-sectional images depict device malapposition at different locations not seen on DSA, with maximum extensions of ~0.35 mm **e** and 0.45 mm **f**. HF-OCT comparison with DSA was repeated for all swine internal maxillary arteries ($n = 16$), following flow-diverting ($n = 16$), and self-expanding intracranial stent implantation ($n = 15$). Scale bars on DSA images are equal to 5.0 mm; HF-OCT scale bars to 1.0 mm.

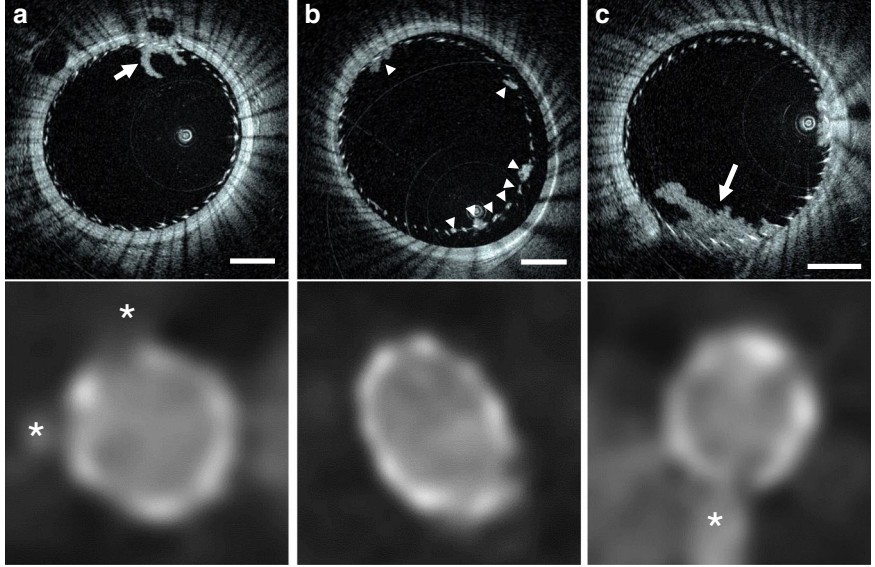

**Fig. 5 Cross-sectional HF-OCT images compared to corresponding CBCT slices. a** A side-branch thrombosis is visible on the HF-OCT image (arrow). **b** Flow-diverter malapposition with a maximum severity of ~400 μm is visible on the HF-OCT image between 1 and 8 o'clock. Small thrombus formations over the flow-diverter struts, with a thickness between ~30 and 220 μm, are indicated by the arrowheads. The presence of thrombus and device malapposition are often undetected on the corresponding cone beam CT images. **c** Thrombosis on the ostium of a large side-branch. HF-OCT comparison with CBCT was repeated for all swine internal maxillary arteries ($n = 16$), following flow-diverting ($n = 16$), and self-expanding intracranial stent implantation ($n = 15$). Scale bars on HF-OCT images are equal to 1 mm. The star symbol (*) on CT images denotes the location of side-branches.

it appears on an image as smearing in the circumferential (rotational) direction, and it causes distortions that may lead to incorrect interpretation and measurements[15]. In the elevated tortuosity of brachial arteries ($n = 16$), HF-OCT data sets had no observed NURD artifacts (Fig. 2). Distortion-free imaging with uniform vessel wall illumination allowed precise visualization of the individual tissue layers including the internal and external elastic laminae, detailed lumen anatomy, and the ostia of two side-branches (Fig. 2b, c).

**Imaging of neurovascular stents and flow-diverters in vivo.** A comparison between digital subtraction angiography (DSA), contrast-enhanced cone beam CT (CBCT), and HF-OCT was obtained in swine internal maxillary arteries (IMAX, $n = 16$), implanted with flow diverting stents (FDS, $n = 16$), and self-expandable intracranial stents (ICS, $n = 15$). Using Fleiss' kappa statistic, agreement between expert image readers ($n = 3$) was evaluated. Values of 0.90 (HF-OCT), 0.67 (CBCT), and 0.49 (DSA) were found for the assessment of acute thrombus

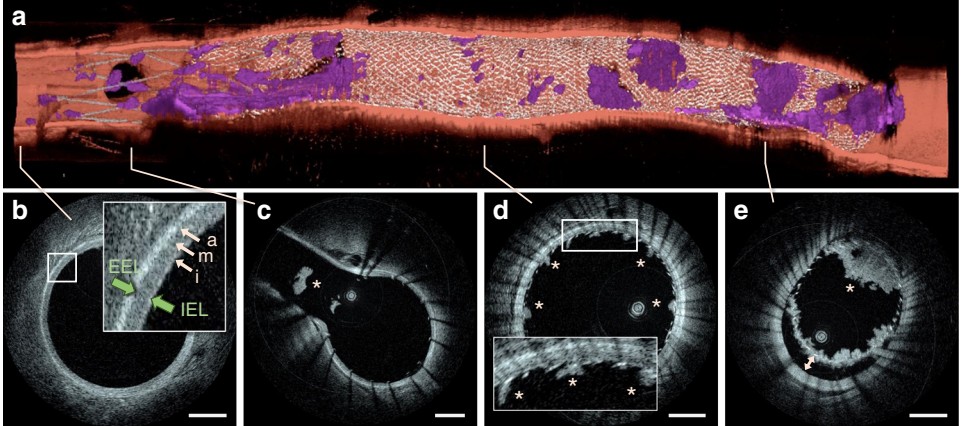

**Fig. 6 HF-OCT volumetric microscopy of an intracranial stent partially overlapping with the distal end of a flow diverting stent. a** Three-dimensional rendering showing significant thrombus accumulation. **b** HF-OCT microscopy depicts the vessel wall microstructure including the individual vessel layers comprising a bright tunica intima, a low-scattering tunica media, and the tunica adventitia (arrows). The internal elastic lamina (IEL) and the external elastic lamina (EEL) are indicated by the green arrows. **c** A thrombus dislodged from the surface of the device floating inside a large branch is visible and denoted by the asterisk. **d** Thrombus formations ranging between 100 and 200 μm in thickness are indicated by the asterisks and distributed over the FDS surface. **e** A semi-occlusive clot formation in correspondence of a significant malapposition (>500 μm) is visible over the proximal end of the FDS. HF-OCT imaging was repeated in a total of $n = 15$ swine internal maxillary arteries with an intracranial stent partially overlapping with a flow-diverting stent. Scale bars are equal to 1.0 mm. Three-dimensional endoscopic rendering color scheme: red, artery wall; purple, clot; silver, flow-diverting stent struts; gray, neurovascular stent struts.

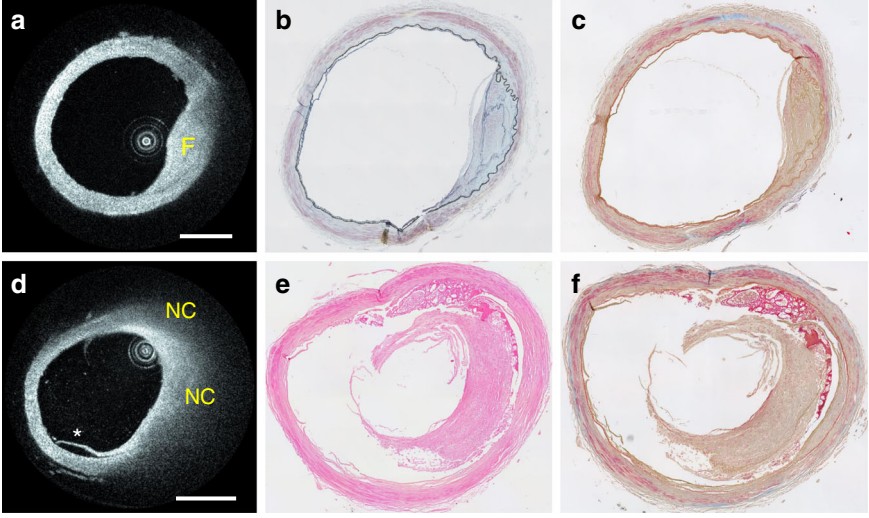

**Fig. 7 Intracranial plaques from an ex vivo segment of the MCA artery. a** HF-OCT imaging of a fibrotic plaque and corresponding trichrome **b** and Movat's staining **c**. Fibrotic tissue is characterized by HF-OCT as a region of homogenous signal, resulting from elevated backscattering and low optical attenuation coefficients. **d** HF-OCT imaging of a necrotic core plaque and corresponding H&E **e** and Movat's staining **f**. A necrotic core plaque is characterized on HF-OCT images as an area with poorly delineated borders followed by an elevated optical attenuation coefficient within an atherosclerotic plaque. The asterisk indicates a vessel wall dissection. From $n = 10$ cadaveric specimens of intracranial arteries, a total of $n = 3$ representative atherosclerotic plaques were identified, processed for histopathology analysis, and compared to HF-OCT imaging. Scale bars are equal to 1.0 mm.

formation along the surface of the FDS. Similarly, values of 0.87 (HF-OCT), 0.67 (CBCT), and 0.18 (DSA) were found for the diagnosis of FDS malapposition. A repeated analysis for the ICS device showed an agreement of 0.81 (HF-OCT), 0.39 (CBCT), and 0.71 (DSA) for thrombus accumulation, and values of 0.78 (HF-OCT), 0.45 (CBCT), and 0.41 (DSA) for malapposition. The relatively good agreement documented on DSA for the presence of thrombus on the ICS was further analyzed. Assuming as ground-truth the observations from HF-OCT, although the reviewers agreed on the DSA studies, more than half of the cases were false-negatives.

Examples of imaging data following FDS implantation are shown in Fig. 3. HF-OCT captured the presence of thrombus accumulation and incomplete device apposition in a comprehensive fashion (Fig. 3a). Similarly, a side-branch occlusion and FDS edge malapposition covered by multiple thrombi are accurately depicted (Fig. 3b, c). Three-dimensional endoscopic data rendering illustrates two perforator-like branches jailed by a FDS (Fig. 3d), including the presence or absence of clots over their ostia (Fig. 3e, f). HF-OCT and corresponding DSA and CBCT data are shown in Figs. 4 and 5. Clots as small as 30 μm (Figs. 4b, c and 5b), malapposition of the device (Fig. 4e, f), and side-branch thrombosis (Fig. 5a, c) are captured by HF-OCT microscopy images, but are unseen on the corresponding DSA and CBCT images. Further examples of the ability of HF-OCT to assess micron-level features of neurovascular devices in vivo are

shown in Fig. 6, including the interaction between two over-lapping devices. Furthermore, a dislodged thrombus in the ostium of a side branch and a partial vessel occlusion as a result of significant thrombosis over a malapposed FDS proximal edge are illustrated in Fig. 6b, e.

**Imaging in large arteries.** To investigate the ability of HF-OCT to image large carotid arteries, additional stenting of swine common carotids was performed in a subgroup of the animals ($n = 5$). The diameter of the stented segments was found to be on average $5.5 \pm 0.3$ mm, with a maximum diameter of 5.9 mm. In all cases, the extended field of view of the Vis-M device provided sufficient illumination to accurately assess the stent-vessel inter-action on a strut level (Supplementary Fig. 2).

**Imaging of intracranial atherosclerosis.** Segments of diseased intracranial arteries ($n = 10$) were obtained at autopsy from patients older than 70 years of age with a history of vascular disease. An expert image reader identified a subset of the arterial segments containing representative examples of the three main plaque types. Representative cases of fibrotic, fibrocalcific, and necrotic core (NC) atherosclerotic disease were found ($n = 3$). Tissue was processed by the means of histopathology techniques and stained using Sudan's black, hematoxylin and eosin (H&E), van Gieson's, Movat Pentachrome, and Trichrome reagents. A vascular pathologist blinded to HF-OCT-imaging results analyzed the stained slices and characterized the plaques and tissue type. Agreement was found in all cases ($n = 3$).

A fibrotic plaque located in the M1 segment of the MCA of a cadaver of 97 years of age is shown in Fig. 7a. Fibrotic tissue was identified on HF-OCT images as a thicker region of the artery wall exhibiting elevated optical backscattering and homogeneous intensity in the near-infrared spectrum[15]. Histopathology assessment categorized this plaque to be mostly composed of fibrotic tissue (Fig. 7b, c). A second plaque containing a NC is illustrated in Fig. 7d. NCs were identified on HF-OCT images as signal-poor areas with poorly delineated boundaries within an atherosclerotic plaque covered by a fibrous cap: necrotic tissue presents a strong optical attenuation in the near-infrared range, resulting in a rapid image intensity signal drop-off with distance into tissue, shadowing the vessel wall region located behind[15]. In agreement with the HF-OCT findings, histopathology classified this plaque as fibrotic containing a NC and underlying media degeneration (Fig. 7e, f). Lastly, a fibrocalcific plaque was identified in the distal segment of an intradural VA from a cadaver of 86 years of age. HF-OCT displays fibrocalcific plaques to contain evidence of fibrotic and calcific tissue, characterized by a signal-poor and heterogeneous region with sharply delineated borders[15]. A fibrocalcific plaque, with a circumferential distribution of 87° and a maximum thickness of ~900 μm is shown in Fig. 8b. Calcific tissue presents low backscattering and attenuation coefficients in the near-infrared range and, as a result, HF-OCT can accurately visualize both its thickness and circumferential distribution. Histopathology assessment categorized this plaque as mostly calcific in agreement with HF-OCT image assessment, with evidence of underlying media degeneration and positive remodeling.

## Discussion

High-resolution IV imaging has the potential to provide sufficient spatial resolution to visualize details of neurovascular arteries and therapeutic devices at a strut level that are unseen by state-of-the-art clinical imaging modalities. Previous research has shown the potential of IV imaging techniques for neurovascular applications[8,21]; however, due to the mechanical characteristics of

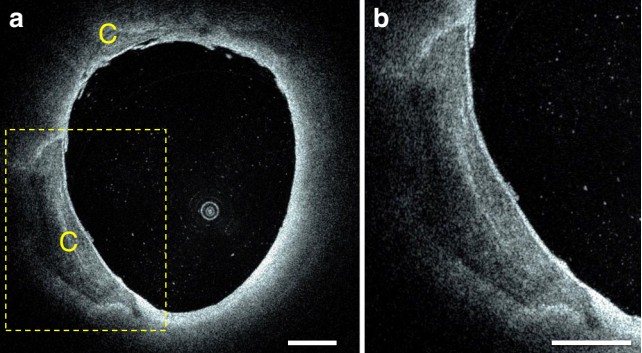

**Fig. 8 Intracranial fibrocalcific plaques in a segment of an intradural vertebral artery. a** Two fibrocalcific plaques are visible in the image. A smaller plaque with a calcium thickness between 100 and 300 μm is located at 11 o'clock. A larger plaque with a calcium maximum thickness of 900 μm is located in the bottom-left quadrant of the image. **b** Magnification showing fine details of the plaque microstructure, including inner and outer boundaries. On HF-OCT, calcific tissue is characterized by a sharply demarcated area with a weaker and heterogeneous signal, resulting from a low optical backscattering and low absorption coefficients. Characterization of HF-OCT versus histopathology for imaging of fibrocalcific plaques was obtained from one intracranial artery specimen. Scale bars are equal to 1 mm.

the existing catheter-based imaging devices, their translation to tortuous cerebrovasculature was unfeasible. The data presented in this study illustrates how the proposed neurovascular HF-OCT system, including a miniaturized, flexible, endovascular-imaging probe enables comprehensive volumetric optical microscopy of intracranial arteries. Using a patient-specific in vitro model of the complete circle of Willis, we have presented translatable contrast injection protocols permitting IV optical imaging at different neurovascular anatomical locations. The optimized protocols established for HF-OCT do not differ from routine practice in neurointerventional surgery for the acquisition of rotational angiography data, and their acquisitions could occur simultaneously. In an animal model, we collected evidence of the suitability of the Vis-M device for use in elevated vascular tortuosity, comparable to the tortuosity conditions encountered in the human ICA where existing IV-imaging devices cannot reliably be used[8]. Volumetric HF-OCT microscopy resulted in a much higher inter-operator agreement for the quantification of intra-luminal clots and for the assessment of the interaction between neurovascular devices and the arterial wall, compared to state-of-the-art imaging modalities such as DSA and CBCT. Furthermore, we illustrated the ability of HF-OCT to visualize the micro-structure of the vessel wall, including characterization of intra-cranial atherosclerosis. Taken together, the results of this study pave the way for the clinical translation of high-resolution endovascular imaging for cerebrovascular use.

A growing body of pre-clinical evidence supports the premise that high-resolution cerebrovascular imaging, with the ability to assess vessel wall disease and neurovascular devices, will have a significant impact on the endovascular treatment of cere-brovascular disease[8–10,21,29,30]. Recently, it has been shown that in a leporine in vivo model the complete apposition of flow-diverting stents is critically important to achieve early and com-plete aneurysm occlusion, but histological evidence of poor apposition was not accurately captured on gold-standard DSA imaging[2]. In a second study, communicating malapposition between the aneurysm neck and flow-diverting devices, captured by IV OCT but unseen on x-ray modalities, was shown to be a predictor of delayed aneurysm occlusion[31]. Similarly, gaps in

intrasaccular device reconstruction of aneurysm necks observed by HF-OCT and undetected on CBCT were shown to correlate with failed aneurysm occlusion[29]. Magnetic resonance is the method of choice for intracranial vessel wall imaging in the context of ischemic stroke and intracranial hemorrhage, however, its accuracy is significantly affected by its limited spatial resolution and voxel size. A recent investigation showed that the higher resolution of IV imaging was able to reveal endothelial injuries, residual thrombus, and ongoing thrombosis of BA perforators at the lesion site following endovascular thrombectomy, in cases where non-invasive angiographic techniques, including CT angiography and magnetic resonance, showed a complete recanalization, free of residual thrombus[23].

The introduced neurovascular HF-OCT system enables volumetric microscopy of the arterial wall at a resolution approaching 10 μm. In this study, we have shown that high-resolution IV imaging can accurately identify intraluminal thrombus and neurovascular devices at a strut level in vivo. These results, in combination with the evidence provided for imaging in tortuous anatomy, suggest that HF-OCT can be used clinically for periprocedural assessment of neurovascular devices so that corrective measures, such as local administration of GP IIb/IIIa inhibitors, angioplasty, or additional stenting, can be deployed. In cases where angiographic findings are ambiguous, its superior resolution allows the visualization of vessel dissections and stent–vessel interactions which non-invasive imaging techniques are unable to capture. An accurate characterization of intracranial atherosclerotic plaque types may inform treatment decisions, as well as enable improved stent sizing, placement, and interaction with perforating arteries and residual stenosis for an improved treatment of intracranial arteries[22,32,33]. Furthermore, cerebrovascular volumetric microscopy by HF-OCT can enable more informed, image guided, personalized antithrombotic management following endovascular treatments[23]. Its enhanced resolution has the ability to study the vessel and aneurysm healing response to an implanted device at an unparalleled level, assessing the tissue thickness over the device surface and vascular remodeling, with precision and accuracy of microscopy techniques in vivo[29,34]. As such, the use of cerebrovascular HF-OCT offers great potential for monitoring device healing in clinical practice, assessing endothelial overgrowth and intimal hyperplasia[35], offering insights for optimal antiplatelet therapy following endovascular treatments[36]. Similarly, in pre-clinical settings, HF-OCT will provide key insights supporting the development and advancement of future generations of neurovascular devices.

With this study, we introduced a neurovascular HF-OCT-imaging system designed to perform cerebrovascular volumetric microscopy. In a combination of in vitro, ex vivo, and in vivo models, we demonstrated the feasibility of a miniaturized HF-OCT endoscopic probe to image tortuous intracranial vascular anatomy, and the ability to visualize fine details of the arterial wall microstructure and its interaction with endovascular devices in high-resolution. The results presented here indicate the potential of HF-OCT technology for use in intracranial arteries and its superiority with respect to state-of-the-art noninvasive imaging modalities to provide additional guidance for neuroendovascular therapeutic procedures.

The ability to inject contrast media and achieve a clear optical window for imaging of neurovascular arteries was investigated in a bench model of the circle of Willis. Substantial efforts were made to reproduce patient circulation, including pulsatile blood flow, physiological flow rates, vessel caliber, and capillary resistance. However, the use of in vitro models may be unable to capture the large spectrum of anatomical variations and disease conditions encountered in patient cerebrovasculature. In this study, a higher viscosity agent was more effective in successfully displacing blood from the

vessel lumen and obtain a clear field of view. To elaborate further, given that a relatively low amount of residual blood can deteriorate HF-OCT image quality (Supplementary Fig. 3B), a higher viscosity media can help to obtain a complete, circumferential, clear field of view more easily, for an optimal image quality acquisition. In clinical practice, other factors such as intracranial artery disease, aging, and consequent reduction of blood flow may permit the use of lower viscosity agents, such as saline, reduced viscosity contrast, or a mixture of both. In these conditions, injection at lower rates may also be possible and provide sufficient clearance for HF-OCT acquisitions. Furthermore, the use of low molecular weight dextran has been successfully demonstrated for coronary imaging[37], safety was investigated for intracranial use[38], and future studies may possibly explore its use for HF-OCT imaging. Similarly, the use of saline or partial dilution with contrast have been demonstrated for IV imaging of coronaries and carotid arteries[39]. As a matter of fact, both saline and dextran are optically transparent media that can be effectively used for HF-OCT imaging. However, as they both present a reduced viscosity[40], injections at higher flow rates will be often required to obtain a fully cleared lumen in vessels with flow rates of 4 ml/s or above. Nevertheless, in more distal vessels or in vessels with impaired flow conditions, the use of saline or dextran may permit good quality imaging as illustrated by previous coronary and peripheral arteries studies[37,41]. Similarly, balloon techniques for proximal occlusion during injection can drastically reduce the blood flow in the target artery and reduce the injection rates of saline, dextran, or contrast. Even though meaningful HF-OCT acquisitions can be acquired through injections limited to 20 ml of contrast or less, saline or dextran may be adopted to avoid or further reduce contrast administration in patients with renal insufficiency and other underlying conditions. In routine clinical workflow, rotational angiography is used for diagnosis and treatment planning of cerebrovascular disease. Given the speed with which the HF-OCT device acquires images (e.g., 50 mm vessel length in 2.5 s), the optimized contrast injection protocol is consistent with that used during rotational angiography allowing for simultaneous acquisitions (rotational angiography and HF-OCT) thereby reducing contrast load.

## Methods

**Neurovascular HF-OCT-imaging system.** In this study, HF-OCT data were collected using prototypes of a neurovascular HF-OCT-imaging system, including an imaging console and an endoscopic-imaging probe, named Vis-M® (Gentuity, Sudbury, MA). HF-OCT describes a long coherence length frequency domain OCT-imaging system specifically designed for use in cerebrovascular anatomy. The term 'high frequency' is used to differentiate the technology introduced in this study from existing commercially available frequency domain OCT (FD-OCT)/OFDI IV imaging solutions[13]. As HF-OCT was designed to address neurovascular imaging requirements, it acquires data at a significantly increased A-line rate and with an extended field of view, generating and collecting signals at higher frequencies. The HF-OCT prototype presented in this study utilizes a central light wavelength of 1300 nm, it has an A-scan line rate of 200 kHz, an axial resolution approaching 10 μm in tissue, and an effective imaging radius of ~7 mm in contrast (diameter of 14 mm), assuming a refractive index $n = 1.45$ for this media[40]. In comparison, commercially available solutions for intracoronary OCT imaging acquire data at 90 kHz, with an imaging radius of 4.8 mm in contrast. The imaging range of the HF-OCT system allows to visualize neurovascular arteries and aneurysms that are up to ~7 mm in diameter, taking into consideration the worst possible scenario where the imaging probe is located in the most eccentric position with respect to the vessel centerline. The Vis-M is a 0.016-inch wire-like catheter device (~400 μm maximum diameter), comprising a distal atraumatic, radiopaque tip. A miniaturized graded index (GRIN) lens is attached to a bend insensitive optical fiber and it is used to focus and collect the backscattered light from the arterial wall, neurovascular devices, and other intraluminal objects. The imaging lens spins within a protective, optically transparent device sheath at a speed of 250 Hz, collecting an equal number of cross-sectional images per second. The device is retracted either manually by the operator or at a pullback speed pre-selected by the user (Fig. 1), and can image arterial segments with a length up to 80 mm in a single acquisition. In case of a 50 mm long pullback sampled in two and a half seconds (i.e., pullback speed of 20 mm/s), the resulting HF-OCT spacing is equal to 80 μm. The primary innovation that enabled device miniaturization and ideal flexibility for

cerebrovascular navigation is the elimination of traditional torque cable solutions adopted by existing IV-imaging technologies, such as IVUS[42] and IV OCT[43]. In the Vis-M construct, a distal rotational control solution comprising a viscous gel is adopted, such that the optical fiber itself can be used to reliably transmit the rotational torque between the system and the distal-imaging lens. From the results presented in this study, it is possible to appreciate how this solution not only allows for a flexible and reduced size endoscopic probe, but also enables a satisfactory image quality collection in conditions showing a severe vascular tortuosity, comparable to the one encountered in human anatomy (Fig. 2). In these situations, traditional torque cable solutions fails, presenting stiffness and safety characteristics not optimized for cerebrovascular use[8].

**Patient-specific vascular model of the circle of Willis**. A patient-specific vascular model including the entire circle of Willis was used to investigate optimal blood clearing protocols at different anatomical locations including the internal carotid, the middle cerebral, the intradural vertebral, and the basilar arteries (Supplementary Fig. 1). Flow sensors and a pressure sensor were used to constantly measure the ICA, MCA, and BA flow rates and monitor the MCA pressure. Compressor clamps were used on the model outlets to adjust peripheral vasculature resistance, and ensure the reproduction of clinically relevant pressure and flow values[44]. Swine blood was maintained at 37 °C and circulated using a pulsatile pump. The resistance of the outlets was adjusted to achieve physiologically representative flow rates through each branch of the model. Blood pressure and flow rate was constantly monitored and adjusted to maintain physiological values of ~250 ml/min in the ICA, 140 ml/min in the MCA, and 160 ml/min in the BA. Importantly, the model includes communicating arteries that have the potential to mix with injected contrast and obscure the acquisition of HF-OCT images.

A pulsatile pump was used to circulate swine blood, and the optimal contrast injection protocols were identified at the location of the ICA, MCA, BA, and VA arteries. For each of these locations, the Vis-M device was delivered to the target anatomy and contrast media infused via a power injector through a 5 F intracranial distal access catheter (Navien, Medtronic Neurovascular, Irvine, CA). Different concentrations of Omnipaque (GE Healthcare, Marlborough, MA) were tested, including 240, 300, and 350 mgI/mL with a viscosity of 3.4, 6.3, and 10.4 cp at 37 °C, respectively. Injections were performed using an automated pump (Medrad Mark 7 Arterion Injection System, Bayer HealthCare, Whippany, NJ) flushing through the 0.058-inch inner lumen of the intracranial distal access catheter at a pressure limit of 300 psi. Starting from 1 ml/s and using increments of 0.5 ml/s, the flow rate was increased until complete clearance was obtained and the optimal flushing protocol for each anatomical location determined. Clearance as a function of the contrast delivery rate was analyzed using a HF-OCT-imaging metric by classifying the results in three categories: blood obscuring the field of view, partial clearance, and complete clearance (Supplementary Fig. 3).

**Animal model and preparation**. A Yorkshire swine model ranging between 40 and 70 kg ($n = 8$) was used in this study in accordance with our University's Institutional Animal Care and Use Committee (IACUC). All procedures were performed under general anesthesia. The animals were pre-anesthetized by a subcuticular injection of glycopyrrolate (0.01 mg/kg). Anesthesia was induced by an intramuscular injection of tiletamine (Telazol, 4.4 mg/kg), ketamine (2.2 mg/kg), and xylazine (2.2 mg/kg) and was maintained through mechanical ventilation of 1–3% isoflurane. During the procedure, the following vital parameters were monitored continuously and recorded: heart and respiratory rate, invasive blood pressure, oxygen saturation, end-tidal $CO_2$, and temperature.

A model of elevated vascular tortuosity was obtained by the means of a flexed forelimb model. This technique is able to provide severe tortuosity in brachial arteries, resulting in radii of curvature similar to the ones encountered in the human ICA[27]. After the surgical exposure of the right femoral artery, a 10 F introducer sheath was inserted for endovascular access and a 5 F Navien™ (Medtronic) was navigated through the proximal segment of the brachial artery allowing the deployment of the Vis-M device to the target anatomy.

The 5 F intracranial distal access catheter was subsequently navigated into the internal maxillary artery (IMAX). FDS ($n = 16$) were deployed bilaterally into the IMAX. A Pipeline FDS (Medtronic) was used in half of the arteries ($n = 8$), and a Surpass FDS (Stryker neurovascular) was implanted in the other half ($n = 8$) by the means of conventional endovascular surgery techniques. Following FDS implantation, an ICS was deployed in each IMAX resulting in a partially overlapping segment. Wingspan stents (Stryker Neurovascular) were implanted in a total of five arteries, Neuroform stents (Stryker Neurovascular) in six arteries, and Solitaire AB stents (Medtronic) in four arteries, for a total of 15 devices. In one single case, no ICS was deployed due to a thrombotic occlusion of the artery at the distal end of the FDS. In addition, a Precise Pro Carotid stent system (Cordis) was implanted in a subset of the animals in the common carotid artery ($n = 5$).

**Acquisition of imaging data**. HF-OCT imaging was performed bilaterally in swine brachial arteries of all animals ($n = 16$). Contrast media (Omnipaque) was injected by the means of a 5 F distal access catheter to displace blood from the arterial lumen. IV HF-OCT images were acquired in all IMAX arteries, following the deployment of FDS and ICS devices, obtaining a total of 31 imaging runs. Sixteen ($n = 16$) of these imaging data set were acquired following the FDS implantation. The additional datasets ($n = 15$) were obtained following the ICS implantation partially overlapping with the FDS. Similarly, DSA, non-subtracted cineangiography, and full-scale, small FOV cone-beam CT images (Philips Healthcare) were acquired for each vessel using standard imaging techniques, obtaining 31 imaging runs for each modality.

**Analysis of neurovascular devices**. Expert image readers ($n = 3$) analyzed the presence of thrombus accumulation along the device and malapposition using a binary outcome metric across the different modalities. Reviewers analyzed a total of 480 images that included angiographic runs, HF-OCT and CBCT cross-sectional images in a blinded fashion. Fleiss' kappa statistic was used to assess the agreement between the different reviewers. A binary scoring system was used to classify all images for the presence or absence of clots and device malapposition. Prior to image quantification, co-registration of the imaging data from the different modalities was obtained using the distal and proximal edges of the FD and ICS as reference markers. Regions of interests (ROI) of 5 mm in length were defined at the proximal and distal edges of all devices. For each ROI, the locations containing the largest clot and the most severe malapposition were matched between DSA, HF-OCT, and the reconstructed CBCT cross-sectional images.

**Ex vivo segments of intracranial artery**. To compare HF-OCT imaging of intracranial atherosclerotic plaques with histopathology, we obtained arterial segments from cadavers older than 70 years of age with a history of smoking and coronary or peripheral artery disease. All specimens were provided by the University of Massachusetts Medical School Division of Translational Anatomy. Per NIH and the University of Massachusetts Medical School IRB guidelines, specimens from deceased individuals are not human subjects research. Consent to use remains for medical research was provided by donors prior to being deceased. The intracranial vasculature of human cadavers was explored ($n = 3$), and multiple arteries appearing to contain atherosclerotic disease were collected ($n = 10$). Segments of the distal internal carotid artery ($n = 2$), MCA ($n = 4$), basilar ($n = 2$), and intradural vertebral ($n = 2$) arteries were harvested. Specimens were traditionally fixated with a 10% formalin solution and subsequently submerged in saline for HF-OCT acquisitions. Multiple data sets capturing volumetric data from the entire length of each artery were obtained. An expert image reader identified regions of interest containing atherosclerotic disease for fibrotic, fibrocalcific, and NC plaques ($n = 3$). Similarly to state-of-art near-infrared IV OCT technology[15], fibrotic plaque tissue was identified by a homogeneous signal with relatively high backscattering; calcific tissue as low attenuation and signal-poor heterogeneous tissue with sharply demarcated borders; necrotic tissue as high-attenuating, resulting in poorly delineated borders, fast signal drop-off, and little or no signal backscattering[15,45]. A visible light indicator emitted by the Vis-M catheter was used to mark each region of interest on the different specimens. The corresponding tissue samples were subsequently embedded in paraffin, sectioned in 5 μm slices, and processed for Sudan's black, H&E, van Gieson's, Movat pentachrome, and trichrome staining.

**Statistical analysis**. Unless otherwise specified, data are presented as mean ± standard deviation. Fleiss' kappa statistics was used to assess and quantify the agreement between the three different image raters classifying DSA, CBCT, and HF-OCT images. Mean and standard deviation statistics were calculated using Microsoft Excel. Fleiss's kappa for inter-rater agreement was calculated using R (Vienna, Austria).

**Three-dimensional HF-OCT renderings**. Cross-sectional images were manually processed and segmented using software ImageJ (NIH). Intraluminal clots, struts of neurovascular devices, and the contour of the vessel wall were identified, manually traced, and labeled. HF-OCT data were rendered in color using the following scheme: red, artery wall; purple, intraluminal clots; silver, metallic struts. The segmented data sets were imported into a volume-rendering DICOM visualization software (OsiriX MD v10.0.2, Pixmeo SARL, Bernex, Switzerland), after a frame-to-frame registration to correct for motion artifacts generated by the mechanical scanning of the catheter (Image J)[46]. Perspective volume rendering techniques using cut surfaces and fly-through methods with manually optimized opacity tables were used for visualization[16].

**Reporting summary**. Further information on research design is available in the Nature Research Reporting Summary linked to this article.

## Data availability
The imaging data that support the findings of this study are available from the corresponding author upon reasonable request.

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

## Acknowledgements

This research was partially supported by the US National Institute of Health (contracts R43 NS100163 and R44 NS100163), by the Massachusetts Life Sciences Center Bits-to-Bytes program, and by Gentuity LLC.

## Author contributions

G.J.U. conceived the study, designed and performed the experiments, reviewed the data, and supervised the project; M.G.M. designed and performed the experiment and analyzed the data; R.M.K., J.C., L.M.P., and B.H.D. performed the experiments and analyzed the data; E.T.L., A.C., and A.L. performed the experiments; S.R. analyzed the data; D.K.L. contributed to design the experiments; M.J.G., designed the experiments, reviewed the data, and supervised the project; A.S.P. designed the experiments and analyzed the data. All authors discussed and edited the manuscript.

## Competing interests

G.J.U., L.M.P., and B.H.D. are employees of Gentuity LLC and hold stocks. M.G.M. is a consultant on a fee-per-hour basis for InNeuroCo In and Stryker Neurovascular. D.K.L. has received research support from Medtronic Neurovascular. M.J.G. has received research support from the National Institutes of Health (NIH), the United States–Israel Binational Science Foundation, Anaconda, Apic Bio, Arsenal Medical, Axovant,

Cerenovus, Ceretrieve, Cook Medical, Galaxy LLC, Gentuity, Imperative Care, InNeuroCo, Insera, Magneto, Microvention, Medtronic Neurovascular, MIVI Neurosciences, Neuravi, Neurogami, Philips Healthcare, Progressive Neuro, Rapid Medical, Route 92 Medical, Stryker Neurovascular, Syntheon, and the Wyss Institute; is a consultant on a fee-per-hour basis for Cerenovus, Imperative Care, Medtronic Neurovascular, Mivi Neurosciences, Phenox, Route 92 Medical, and Stryker Neurovascular; holds stock in Imperative Care, InNeuroCo, and Neurogami. A.S.P. has been a proctor on a fee-per-hour basis for Stryker Neurovascular, and Cerenovus; has research grants from Cerenovus, Medtronic Neurovascular, and Stryker Neurovascular; holds stock in InNeuroCo, and NTI. The remaining authors declare no competing interests.
