## [Peer Review File · Nature Communications]

Reviewers' comments:

Reviewer #1 (Remarks to the Author):

Current high-resolution cerebrovascular imaging relies on 3-dimensional rotational angiography (3DRA) and cone-beam computed tomographic angiography (CB-CTA). These acquisitions require significant additional radiation exposure and contrast injection, in addition to the conventional digital subtraction imaging. Both 3DRA and CB-CTA are limited in resolution to 50-100 microns under optimal conditions, and are often limited by patient movement (both involuntary as well as respiration-related and arterial pulsation-related). Since 3DRA and CB-CTA acquisitions require 5-20 seconds duration for acquisition, they result in time-averaged results, and often a motion-artifact and noise averaged over the acquisition that limits its utility in sub-optimal conditions. The clinical relevance of high-resolution imaging such as afforded by the technique described by the authors has become increasingly recognized to be critical to achieve optimal clinical outcomes. One example is the need to recognize the exact pathological features of intracranial atherosclerosis, another is to determine stent-vessel wall apposition, and yet a third is to obtain additional information on embolic material in large vessel occlusion acute stroke cases to better guide mechanical thrombectomy attempts.

With that background the authors' work and the development of HF-OCT comes at an opportune time, and provides unsurpassed spatial resolution in a short period of time, that does not require additional radiation exposure to the patient. The careful characterization of this new technology by the authors should be lauded. It is hoped that future studies from the authors' group and others should further elucidate the advantages of this approach and validate its clinical utility in specific sub-conditions, and possibly expand its utility to beyond the cerebrovascular system.

Question to the authors:

1-They describe using contrast material to clear the vessel to enable HF-OCT function (line 108); did they compare to saline solution infusion or another higher viscosity solution such as Dextran? In certain patients with renal insufficiency, it would be desirable to avoid additional contrast administration. The authors touch on this point in the Discussion (line 281).

2-Would it be possible to include an elastic compliant proximal balloon for temporary occlusion during the injection or at least to slow blood flow down and minimize contrast needed for injection to enable HF-OCT functionality?

3-Could the authors comment on how sensitive the accuracy of HF-OCT is to complete clearance of the lumen during acquisition? How much does the image deteriorate if the blood clearance is imperfect or non-uniform? It is not obvious from the data presented and the readers would appreciate a discussion. Although the authors describe it on line 310, the effective clearance parameters may be more or less challenging intracranially where laminar flow and streaming may provide a radially non-uniform contrast distribution; please discuss.

4-Could you please discuss if any vibration or mechanical forces are exerted by the wire during the rotation process and its possible effect on dislodging or disturbing any thrombi that may be adherent to the endovascular device or atherosclerotic plaque.

Minor:

1-Spelling line 132, "imaging aith angiography" should be "imaging with angiography"

Adel M. Malek, MD, PhD

Reviewer #2 (Remarks to the Author):

The authors report an OCT imaging catheter for acquiring volumetric microscopy data from cerebral vascular anatomy. This technique, termed high-frequency optical coherence tomography (HF-OCT), provides a clear visualization of the cerebral arterial microstructure. The key technical advancement is that the presented device can enter the tortuous cerebrovascular structure. Current intravascular OCT, widely used in coronary artery imaging, lacks the flexibility and the size is not suitable for cerebral artery imaging. The small size (0.016-inch wire-like catheter) and the flexibility overcome the existing limitations. The images of ex vivo and in vivo models of cerebrovascular arteries with neuroendovascular devices are promising. Here are my comments and questions:

1. Several references, which are original or relevant, should be included. Here are some suggestions:
 - Yun et al., Comprehensive volumetric optical microscopy in vivo, Nature Medicine, 2006
 - Nam et al., Automated detection of vessel lumen and stent struts in intravascular optical coherence tomography to evaluate stent apposition and neointimal coverage, Medical Physics, 2016
2. Why is this technique is termed high-frequency OCT? What does "high-frequency" mean?
3. Much more information about the catheter and the system must be added to the manuscript. Because the most important technological advancement of the presented work is the miniaturization of the catheter and the high-speed imaging system. How is the catheter composed? What kind of fiber is used? Was GRIN lens or ball lens used? What is the size of the fiber? Was torque coil used? How can the torque be translated with such a small size? What is the sheath material and size? A-line rate of the system? How was the large field-of-view obtained? And others.
4. Page 6, does "(angiography)" mean DSA?

Reviewer #3 (Remarks to the Author):

In this work, the authors present a set of impressive imaging studies using what appears to be a company/proprietary product termed HF-OCT for cerebrovascular imaging. The field would find value in seeing the current capabilities of the OCT technology in this space, and I congratulate the authors on these high-quality imaging results.

However, the work is presented as the introduction of a new imaging platform HF-OCT, yet inadequately references and describes prior works demonstrating the use of OCT in this context (see list of some of the missing references below), and is entirely vague on the unique features or capabilities of the HF-OCT platform. The specific features or advances of HF-OCT are never described.

Continuing on this later point, the HF-OCT platform is never described even at a high level. It is unclear what HF means, i.e., what is high-frequency. High RF frequency? High rotational frequency? The A-line rate is never provided. The imaging range that the system can support is not described (yet achieving larger imaging range is claimed). The frame rate is somewhat buried in the methods, but pullback speed, frame pitch, etc are undefined. Even the wavelength at which the imaging is being done is not described. Is the advance in the OCT portion itself, or in the catheter? Or is it both? Given that the central thesis of this work is that prior OCT methods are technically inadequate to address the cerebral vasculature, it is required that the technical innovations in HF-OCT that overcome these challenges be described. Therefore, I believe that the manuscript in its current form, the manuscript lacks clarity and transparency required for publication in Nature Communications. It is not acceptable, for example, that after reading the

manuscript, I cannot point to single technical specification of the HF-OCT that is improved over existing OCT methods.

Alternatively, if the manuscript focused instead on demonstrating the use of OCT in vessels/sites/applications that had previously been impossible to image, or in clinical settings where prior reports were all preclinical, or otherwise was based on the first demonstration of a new capability, then these novel demonstrations could serve to differentiate the work from prior reports (rather than statements that the HF-OCT platform is superior). However, it is difficult to judge from the current manuscript which of the imaging results are entirely novel and enabled (rather than enhanced) by the HF-OCT.

Examples of some of the work showing OCT being used for similar applications as described in the manuscript but not referenced:

Attizzani GF, Jones MR, Given CA, Brooks WH, Bezerra HG, Costa MA. Frequency-domain optical coherence tomography assessment of very late vascular response after carotid stent implantation. *J Vasc Surg.* 2013; 58:201–204

Jones MR, Attizzani GF, Given CA, Brooks WH, Costa MA, Bezerra HG. Intravascular frequency-domain optical coherence tomography assessment of atherosclerosis and stent-vessel interactions in human carotid arteries. *AJNR Am J Neuroradiol.* 2012

Reviewer #1 (Remarks to the Author):

Current high-resolution cerebrovascular imaging relies on 3-dimensional rotational angiography (3DRA) and cone-beam computed tomographic angiography (CB-CTA). These acquisitions require significant additional radiation exposure and contrast injection, in addition to the conventional digital subtraction imaging. Both 3DRA and CB-CTA are limited in resolution to 50-100 microns under optimal conditions and are often limited by patient movement (both involuntary as well as respiration-related and arterial pulsation-related). Since 3DRA and CB-CTA acquisitions require 5-20 seconds duration for acquisition, they result in time-averaged results, and often a motion-artifact and noise averaged over the acquisition that limits its utility in sub-optimal conditions. The clinical relevance of high-resolution imaging such as afforded by the technique described by the authors has become increasingly recognized to be critical to achieve optimal clinical outcomes. One example is the need to recognize the exact pathological features of intracranial atherosclerosis, another is to determine stent-vessel wall apposition, and yet a third is to obtain additional information on embolic material in large vessel occlusion acute stroke cases to better guide mechanical thrombectomy attempts.

With that background the authors' work and the development of HF-OCT comes at an opportune time and provides unsurpassed spatial resolution in a short period of time, that does not require additional radiation exposure to the patient. The careful characterization of this new technology by the authors should be lauded. It is hoped that future studies from the authors' group and others should further elucidate the advantages of this approach and validate its clinical utility in specific sub-conditions, and possible expand its utility to beyond the cerebrovascular system.

We appreciate the kind words regarding the significance of the work described in this manuscript.

Question to the authors:

1. They describe using contrast material to clear the vessel to enable HF-OCT function (line 108); did they compare to saline solution infusion or another higher viscosity solution such as Dextran? In certain patients with renal insufficiency, it would be desirable to avoid additional contrast administration. The authors touch on this point in the Discussion (line 281).

We agree with the reviewer's remark that the use of dextran or saline to reduce amount of contrast administered to patient with renal insufficiency is indeed of interest for HF-OCT IV imaging. Both dextran and saline are transparent media that can be successfully used for optical imaging. We have fully addressed this important remark in the revised manuscript (page 11-12). We further emphasize that given the image acquisition rate allowing HF-OCT imaging of 50mm of vessel within 2.5s, the optimized contrast injection protocol described is consistent with those used for acquiring rotational angiography. Thus, simultaneous acquisition of HF-OCT and rotational angiography may help to reduce contrast load in this patient population.

2. Would it be possible to include an elastic compliant proximal balloon for temporary occlusion during the injection or at least to slow blood flow down and minimize contrast needed for injection to enable HF-OCT functionality?

This is another noteworthy point. As indicated by the reviewer, the use of proximal balloons may indeed drastically reduce the blood flow, the amount, and flush rate required for an optimal contrast injection, allowing the use of other media such as dextran or saline, avoiding additional contrast administration. We addressed this comment in the revised manuscript (page 11-12).

3. Could the authors comment on how sensitive the accuracy of HF-OCT is to complete clearance of the lumen during acquisition? How much does the image deteriorate if the blood clearance is imperfect or non-uniform? It is not obvious from the data presented and the readers would appreciate a discussion. Although the authors describe it on line 310, the effective clearance parameters may be more or less challenging intracranially where laminar flow and streaming may provide a radially non-uniform contrast distribution; please discuss.

As requested by the reviewer, we have fully addressed this point in the revised manuscript (page 11-12).

4. Could you please discuss if any vibration or mechanical forces are exerted by the wire during the rotation process and its possible effect on dislodging or disturbing any thrombi that may be adherent to the endovascular device or atherosclerotic plaque.

Interesting remark. Particular attention was given to this and other safety aspects during device development. In the here described endovascular imaging device, a miniaturized optical fiber with a glass of less than 100 μm is contained and rotating inside a plastic sheath. As such, the vessel wall is never exposed to the rotating element, it is protected by the plastic sheath, and no vibration or mechanical forces are exerted on the vessel wall by the distal portion of the imaging probe. The safety of 'rotational' intravascular imaging techniques has been studied in detail in thousands of patients with excellent results (). The construct of the device presented in this manuscript being less than half the size of commercially available designs and the absence of a torque cable can only make this device safer. This point is now explicitly addressed on the revised manuscript (page 4 and page 12).

Minor: Spelling line 132, "imaging aith angiography" should be "imaging with angiography"

This typo has been corrected in the revised manuscript.

Reviewer #2 (Remarks to the Author)

The authors report an OCT imaging catheter for acquiring volumetric microscopy data from cerebral vascular anatomy. This technique, termed high-frequency optical coherence tomography (HF-OCT), provides a clear visualization of the cerebral arterial microstructure. The key technical advancement is that the presented device can enter the tortuous cerebrovascular structure. Current intravascular OCT, widely used in coronary artery imaging, lacks the flexibility and the size is not suitable for cerebral artery imaging. The small size (0.016-inch wire-like catheter) and the flexibility overcome the existing limitations. The images of ex vivo and in vivo models of cerebrovascular arteries with neuroendovascular devices are promising.

We would like to thank the reviewer for the positive review, for the important remarks regarding the limitations of current IV OCT imaging modalities and their inadequate characteristics for cerebrovascular imaging. We appreciate the chance to improve the originally submitted manuscript. We have addressed all comments and remarks as described below.

Here are my comments and questions:

5. Several references, which are original or relevant, should be included. Here are some suggestions:
 - Yun et al., Comprehensive volumetric optical microscopy in vivo, Nature Medicine, 2006
 - Nam et al., Automated detection of vessel lumen and stent struts in intravascular optical coherence tomography to evaluate stent apposition and neointimal coverage, Medical Physics, 2016

As recommended by the reviewer, we have added these two additional references plus other relevant references (e.g., Tearney, G. J. et al. Three-dimensional coronary artery microscopy by intracoronary optical frequency domain imaging. JACC. Cardiovascular imaging 1, 752-761, doi:10.1016/j.jcmg.2008.06.007 (2008)) to the introduction section of the manuscript.

6. Why is this technique is termed high-frequency OCT? What does "high-frequency" mean?

We appreciate the chance to improve our manuscript and enhance clarity for future readers. In combination with the comment below (and the remarks of reviewer #3), we have decided to provide a much more detailed description of the imaging system and device used in this study and to elucidate the meaning of the 'high-frequency' term.

7. Much more information about the catheter and the system must be added to the manuscript. Because the most important technological advancement of the presented work is the miniaturization of the catheter and the high-speed imaging system. How is the catheter composed? What kind of fiber is used? Was GRIN lens or ball lens used? What is the size of the fiber? Was torque coil used? How can the torque be translated with such a small size? What is the sheath material and size? A-line rate of the system? How was the large field-of-view obtained? And others.

We have taken in deep consideration this comment as well as the remark made by reviewer number 3. To fulfill the reviewers' request, we have provided a much more detailed description about the catheter and system to the manuscript, as demanded (pages 4-5 and 11-12). With this additional description, we have now included a large number of details providing a complete device description for the readers of this article as well as future clinical users. We appreciate the chance to improve the original submission.

8. Page 6, does "(angiography)" mean DSA?

Correct. We apologize for the confusion about the term 'angiography'. To improve clarity, we replaced the term 'angiography' with the 'DSA' in the revised manuscript (page 6).

Reviewer #3 (Remarks to the Author):

In this work, the authors present a set of impressive imaging studies using what appears to be a company/proprietary product termed HF-OCT for cerebrovascular imaging. The field would find value in seeing the current capabilities of the OCT technology in this space, and I congratulate the authors on these high-quality imaging results.

We are grateful to the reviewer for the enthusiastic comment.

However, the work is presented as the introduction of a new imaging platform HF-OCT, yet inadequately references and describes prior works demonstrating the use of OCT in this context (see list of some of the missing references below), and is entirely vague on the unique features or capabilities of the HF-OCT platform. The specific features or advances of HF-OCT are never described. Continuing on this later point, the HF-OCT platform is never described even at a high level. It is unclear what HF means, i.e., what is high frequency. High RF frequency? High rotational frequency? The A-line rate is never provided. The imaging range that the system can support is not described (yet achieving larger imaging range is claimed). The frame rate is somewhat buried in the methods, but pullback speed, frame pitch, etc. are undefined. Even the wavelength at which the imaging is being done is not described. Is the advance in the OCT portion itself, or in the catheter? Or is it both? Given that the central thesis of this work is that prior OCT methods are technically inadequate to address the cerebral vasculature, it is required that the technical innovations in HF-OCT that overcome these challenges be described. Therefore, I believe that the manuscript in its current form, the manuscript lacks clarity and transparency required for publication in Nature Communications. It is not acceptable, for example, that after reading the manuscript, I cannot point to single technical specification of the HF-OCT that is improved over existing OCT methods. Alternatively, if the manuscript focused instead on demonstrating the use of OCT in vessels/sites/applications that had previously been impossible to image, or in clinical settings where prior reports were all preclinical, or otherwise was based on the first demonstration of a new capability, then these novel demonstrations could serve to differentiate the work from prior reports (rather than statements that the HF-OCT platform is superior). However, it is difficult to judge from the current manuscript which of the imaging results are entirely novel and enabled (rather than enhanced) by the HF-OCT.

We have taken in deep consideration this comment from the reviewer and we appreciate the opportunity for improving our original submission. To satisfy this comment (and the remark from Reviewer nr. 2), we have decided to add a specific section to paper 'materials and methods', providing the reader with the details demanded above (pages 4-5 and 11-12). As ask for, we have described the meaning of the term 'high-frequency', A-line rate, imaging range, and the wavelength of the system. We additionally provided

the pullback speed, the frame pitch of collected data, as well as a description of the critical advances of this imaging technology for achieving the desired specs for cerebrovascular imaging as recommended by the reviewer. Furthermore, to clarify the latter remark about 'vessels/site/applications', we need to stress the fact that we have selected for our in vivo evaluation the swine brachial artery model. This model was specifically selected as it has similar tortuosity but smaller diameter (see reference and image below) to the one encountered by human ICA resulting in a worst-case scenario. In this anatomy, existing IV imaging solutions designed for the coronaries fails, as they are unable to deal with this elevated level of tortuosity (as also noted by the second reviewer). As such, the results from this paper do illustrate the ability of the proposed HF-OCT technology to image in vivo elevated tortuosity which is impossible to image with currently available IV imaging techniques. As this is another very important remark made by the reviewer, we have decided to better clarify this point throughout the manuscript (page 6).

Examples of some of the work showing OCT being used for similar applications as described in the manuscript but not referenced:

- Attizzani GF, Jones MR, Given CA, Brooks WH, Bezerra HG, Costa MA. Frequency-domain optical coherence tomography assessment of very late vascular response after carotid stent implantation. *J Vasc Surg*. 2013; 58:201–204
- Jones MR, Attizzani GF, Given CA, Brooks WH, Costa MA, Bezerra HG. Intravascular frequency-domain optical coherence tomography assessment of atherosclerosis and stent-vessel interactions in human carotid arteries. *AJNR Am J Neuroradiol*. 2012

As requested (also see reviewer #2, comment #5), we have improved the use of references in the introduction section, added the above-mentioned references, plus some additional ones (e.g., Given, CA, Ramsey CN, Attizzani GF, Jones MR, Brooks WH, Bezerra HG, Costa MA. Optical coherence tomography of the intracranial vasculature and Wingspan stent in a patient. *J Neurointerv Surg* 7, e22, doi:10.1136/neurintsurg-2014-011114.rep (2015)).

REVIEWERS' COMMENTS:

Reviewer #2 (Remarks to the Author):

There is still one very critical question. What is the distal rotational control solution adopted in this study? This is the most important advance of the presented work over previous works. A detailed description of the distal rotational control solution should be added to the manuscript.

Reviewer #3 (Remarks to the Author):

The authors have significantly improved the manuscript. Changes address the primary concerns raised in my initial review. While better, I still encourage the authors to further adjust their language to accurately position this work. I believe these results are important contributions to our understanding of how OCT could be used to image cerebral vasculature. However, the instrumentation and technology that is used are engineering modifications to existing approaches. This is not a criticism, except that the language of the title and manuscript (see for example the abstract) does not accurately communicate this, but instead the messaging is that this work describes a new imaging modality. I think this will create confusion in both the engineering and clinical communities, and further it is not necessary. Instead, I suggest the authors emphasize the results that have been achieved, rather than what I believe is misleading language that positions the instrument as a novel approach to intravascular imaging. See the following sentence in abstract as an example: "We present a novel fiber-optic imaging technology termed high-frequency optical coherence tomography (HF-OCT) designed to rapidly acquire volumetric microscopy data at a resolution approaching 10 microns in highly tortuous cerebrovascular anatomies." Many readers will be misled by this sentence - the HF-OCT is not novel. It is the same intravascular OCT platform that has been used for 10+ years, with some optimizations that enable it to be better used in specific settings.

Following editorial office and reviewers' comments, we have further revised our manuscript. New and revised text is highlighted using red color wherever possible.

REVIEWERS' COMMENTS:

Reviewer #2 (Remarks to the Author):

There is still one very critical question. What is the distal rotational control solution adopted in this study? This is the most important advance of the presented work over previous works. A detailed description of the distal rotational control solution should be added to the manuscript.

As requested by reviewer #2, we have added to the revised manuscript a description regarding rotational control solution.

Reviewer #3 (Remarks to the Author):

The authors have significantly improved the manuscript. Changes address the primary concerns raised in my initial review. While better, I still encourage the authors to further adjust their language to accurately position this work. I believe these results are important contributions to our understanding of how OCT could be used to image cerebral vasculature. However, the instrumentation and technology that is used are engineering modifications to existing approaches. This is not a criticism, except that the language of the title and manuscript (see for example the abstract) does not accurately communicate this, but instead the messaging is that this work describes a new imaging modality. I think this will create confusion in both the engineering and clinical communities, and further it is not necessary. Instead, I suggest the authors emphasize the results that have been achieved, rather than what I believe is misleading language that positions the instrument as a novel approach to intravascular imaging. See the following sentence in abstract as an example" "We present a novel fiber-optic imaging technology termed high-frequency optical coherence tomography (HF-OCT) designed to rapidly acquire volumetric microscopy data at a resolution approaching 10 microns in highly tortuous cerebrovascular anatomies." Many readers will be misled by this sentence - the HF-OCT is not novel. It is the same intravascular OCT platform that has been used for 10+ years, with some optimizations that enable it to be better used in specific settings.

Following reviewer #3 recommendation, we have revised the language throughout the whole manuscript, including manuscript title and abstract. We appreciate the chance to further revise, improve, and clarify the manuscript.